# Comparison of Pedotransfer Functions for Determination of Saturated Hydraulic Conductivity for Highly Eroded Loess Soil

**Agnieszka Petryk** [1,*] **, Edyta Kruk** [2] **, Marek Ryczek** [2] **and Lenka Lackóová** [3]

1 Department of Urban Management, Institute of Spatial Management and City Studies, College of Public Economy and Administration, Cracow University of Economics, Rakowicka 27 St, 31-510 Kraków, Poland

2 Department of Land Reclamation and Environmental Development, Faculty of Environmental Engineering and Land Surveying, Agriculture University of Krakow, Mickiewicza 24/28 St, 30-059 Krakow, Poland

3 Institute of Landscape Engineering, Faculty of Horticulture and Landscape Engineering, Slovak University of Agriculture in Nitra, Hospodárska, 794901 Nitra, Slovakia

* Correspondence: petryka@uek.krakow.pl; Tel.: +48-122-937-420

**Abstract:** Saturated hydraulic conductivity is one of the most essential soil parameters, influencing surface runoff and water erosion formation. Both field and laboratory methods of measurement of this property are time or cost-consuming. On the other hand, empirical methods are very easy, quick and costless. The aim of the work was to compare 15 pedotransfer models and determination of their usefulness for assessment of saturated hydraulic conductivity for highly eroded loess soil. The mean values obtained by use of the analyzed functions highly fluctuated between $2.00 \cdot 10^{-3}$ and $4.05 \cdot 10^{0}$ m·day$^{-1}$. The results of calculations were compared within them and with the values obtained by the field method. The function that was the best comparable with the field method were the ones proposed by Kazeny-Carman, based on void ratio and specific area, and by Zauuerbrej, based on total porosity and effective diameter $d_{20}$. In turn, the functions that completely differed with the field method were the ones proposed by Seelheim, based on effective diameter $d_{50}$ and by Furnival and Wilson, based on bulk density, organic matter, clay and silt content. The obtained results are very important for analysis among others water erosion on loess soil.

**Keywords:** saturated hydraulic conductivity; pedotransfer functions; loess soil

## 1. Introduction

Water erosion, in addition to drought, flooding, salinity, contamination by heavy metals, waste disposaland peat-bogs decay, is one of the forms of earth surface degradation. Ref. [1] carried out simulations of water erosion intensity caused by various land use scenarios in the highly eroded mountain Mątny stream basin located in the Western Carpathians. Ref. [2] analyzed soil erosion in China between 1980 and 2010, incorporated with landform, slope, vegetation coverage, land use and remote sensing images. This process take place especially in loess areas all over the world [3–7]. Ref. [8] carried out a laboratory experiment devoted to the influence of drying and rewetting cycles on respiratory processes in organic soil. Ref. [9] carried out investigations on salinity problems connected with landfill of the Cracow Soda Plant. Ref. [10] presented the state of the art connected with drivers, indicators and monitoring, modeling and mapping methods for salinity of soil. Refs. [11,12] elaborated the dependence of salinity and sodicity levels on irrigation water quality, using a numerical approach. Ref. [13] investigated the influence of soil salinity on microorganisms and respiratory responses. Ref. [14] presented the optimization method in optimizing the parameters of the salinity stress reduction function, establishing the root-water-uptake model and simulating soil water flow under the salinity stress condition. Ref. [15] studied the concentrations of heavy metals in water, sediment, zooplankton and fish in the coastal waters of Kalpakkam, near a nuclear power plant. Ref. [16] investigated the significance of halophytes in conditions of high salinity and their role in the process of phytoremediation

of heavy metals. Ref. [17] examined the water retention ability of chosen industrial wastes taken from landfills. Ref. [18] determined physical, hydrophysical and chemical properties of the upper layer of peat soil on post-extracted areas. One of the soil properties regarded in water erosion evaluation is saturated hydraulic conductivity [19–21]. It is taken into account as a criteria for hydrologic group identification for CN parameter and maximum potential basin retention determination [22]. It is one of the parameters for water transport models in the unsaturated zone [23] and depends mainly on: texture, bulk density and organic matter content [7,24]. This soil property is characterized by particular high spatial variability [25–29]. There are many methods for saturated hydraulic conductivity determination. In general, the methods can be classified as: laboratory, field and empirical ones [30]. The laboratory and fields methods are the most accurate, but they are time and cost consuming. In turn, the empirical methods are quick and easy, as usually they require only knowledge of the grain size distribution curve and some physical properties of soil and water (for example, total porosity and water specific density) [30–36]. They are grouped in three categories. The simplest are based only on some effective diameters, taken from grain size distribution curve. The second ones, apart from effective diameters, take into account chosen soil physical properties, most often porosity. The third ones are based apart above properties, on physical properties of water, such as: specific density or viscosity [37]. In the literature, the methods are most commonly reported as the pedotransfer functions (PTF) [24,38–45]. The pedotranfer functions can be regarded as wider term than empirical functions. This term was proposed for the first time by [46], although such an approach depends on the estimation of soil properties from other more easily measurable soil properties [47]. This had been known since the early 20th century [48]. The easy measurable parameters are called predictors. They are: sand, silt and clay fractions content [49–51], organic matter or organic carbon content and bulk density [12,50,52–55]. Explaining parameters are most often: hydraulic parameters (hydraulic conductivity, water retention), solute transport parameters (preferential flow, solute transport), thermal parameters (thermal conductivity) and biogeochemical parameters (adsorption isotherm, carbon stocks) [47].

## 2. Material and Methods

### 2.1. Pedotransfer Functions

In this work there were used 15 pedotransfer functions for determination of saturated hydraulic conductivity:

**Method 1—Hazen [36]:**

$$K_s = c \cdot d_{10}^2 [\text{m} \cdot \text{day}^{-1}]$$

where:

$K_s$—saturated hydraulic conductivity [m·day$^{-1}$]
$d_{10}$—effective grain size, soil particle diameter [mm] such that 10% of all particles are finer by weight.
$c$—a constant that varies from 1.0 to 1.5 if $K_s$ is expressed in cm·s$^{-1}$ in original method proposed by Hazen; in the work it was taken according to Lange as: $c = [400 + 40 \cdot (n - 26)]$, where $n$ is total porosity (%).

**Method 2—Hazen—Tkaczukowa [56]:**

$$K_s = 864 \cdot \frac{0.0093}{a^2} \cdot d_{10}^2 [\text{m} \cdot \text{day}^{-1}]$$

where:

$a$—content of particles of diameter $d < 0.001$ mm [-],
$d_{10}$—as above.

**Method 3—USBR** [57]:

$$K_s = 86{,}400 \cdot 0.0036 \, d_{20}^{2,3} \, [\text{m} \cdot \text{day}^{-1}]$$

where:

$d_{20}$—effective grain size soil particle diameter [mm] such that 20% of all particles are finer by weight.

**Method 4—Saxton et al.** [44,58]:

$$K_s = e^{12.012 - 0.0755 \cdot (S_i) + (-3.895 + 0.0367 \cdot (S_i) - 0.1103 \cdot (C)) + \frac{0.00087546 \cdot (C)^2}{\theta_s}} \, [\text{m} \cdot \text{day}^{-1}]$$

where:

$C$—clay fraction content (<0.002 mm) [%],
$S_i$—silt fraction content (0.05–0.002 mm) [%],
$\theta_s$—saturated soil moisture [$\text{m}^3 \cdot \text{m}^{-3}$], calculated as: $\theta_s = 0.332 - 0.0007251 \cdot S + 0.1276 \cdot log(C)$.

**Method 5—Kozeny—Carman** [31]:

$$K_s = \left(\frac{\gamma}{\mu}\right) \cdot \left(\frac{1}{C_{KC} \cdot S_0^2}\right) \cdot \left(\frac{e^3}{1+e}\right) \, [\text{m} \cdot \text{day}^{-1}]$$

where:

$\gamma$—specific density of water [$\text{Mg} \cdot \text{m}^{-3}$],
$\mu$—dynamic liquid viscosity coefficient [$\text{m} \cdot \text{s}^{-2}$],
$e$—void ratio [-],
$S_0$—specific area [$\text{cm}^{-1}$], in the work it was measured by gravimetric method (glycerine as absorber)
$C_{KC}$—Kozeny-Carman constant, taken most often as 5.

**Method 6—Krűger** [37]:

$$K_s = 322 \cdot \frac{n}{(1-n)^2} \cdot d_e^2 \, [\text{m} \cdot \text{day}^{-1}]$$

where:

$n$—total porosity (-),
$d_e$—effective diameter (mm) calculated as: $d_e = \frac{100}{\sum_1^N \frac{a_i}{d_i}}$, where: $N$—number of fraction,

$a_i$—percentage of following fractions in texture, $d_i$—grain diameter within following fractions from 1 to $N$ (mm), calculated as: $d_i = \frac{d_y + d_x}{2}$, where: $d_y$ and $d_x$—lower and upper diameter of following fractions from 1 to $N$.

**Method 7—Terzaghi** [34,59]:

$$K_s = \frac{C}{\eta} \cdot \left(\frac{n - 0.13}{\sqrt[3]{1-n}}\right)^2 \cdot d_{10}^2 \cdot (1 + 0.034 \cdot t) \, [\text{m} \cdot \text{day}^{-1}]$$

where:

$C$—coefficient depending on shape of particles, equal to 10.48 for round and 6.02 for sharp edge particles [-],
$h$—viscosity coefficient [$\text{Pa} \cdot \text{s}$],
$n$—as above,
$d_{10}$—as above,
$t$—temperature of water [$°\text{C}$]

**Method 8—Chapuis [32]:**

$$K_s = 864 \cdot 2.4622 \cdot d_{10}^2 \cdot \left( \frac{e^3}{1+e} \right)^{0.7825} \; [\text{m} \cdot \text{day}^{-1}]$$

where:

$e$—void ratio [-],
$d_{10}$—as above.

**Method 9—Seelheim [60]:**

$$K_s = 864 \cdot 0.357 \cdot d_{50}^2 \; [\text{m} \cdot \text{day}^{-1}]$$

where:

$d_{50}$—effective diameter[mm], such that 50% of all particles are finer by weight.

**Method 10—NAVFAC [32,61]:**

$$K_s = 864 \cdot 10^{1.291 \cdot e - 0.6435} \cdot d_{10}{}^{10^{0.5504 - 0.2937 \cdot e}} \; [\text{m} \cdot \text{day}^{-1}]$$

where:

$e$—void ratio [-],
$d_{10}$—as above.

**Method 11—Sauerbrej [57]:**

$$K_s = \beta \cdot \frac{n^2}{(1-n)^2} \cdot d_{20}^2 \; [\text{m} \cdot \text{day}^{-1}]$$

where:

$\beta$—empirical coefficient [-] depending on dimension and grain size homogeneity, it takes value between 1150 and 3010 (usually 2880–3010, in the work it was taken as 2945,
$d_{20}$—as above,
$n$—as above.

**Method 12—Slichter [62]:**

$$K_s = 86,400 \cdot 8.83 \cdot d_{10}^2 \cdot \frac{1}{\mu} \cdot m \; [\text{m} \cdot \text{day}^{-1}]$$

where:

$d_{10}$—as above,
$m$—coefficient depending on porosity, $m = 0.0039 \cdot n - 0.0012$, where n is total porosity [$-$],
$\mu$—dynamic viscosity of water [Pa·s].

**Method 13—Furnival and Wilson [63]:**

$$K_s = 9.5 - 1.471 \cdot BD^2 - 0.688 \cdot OM + 0.0369 \cdot OM^2 - 0.332 \cdot \ln(C + S_i) \; [\text{m} \cdot \text{day}^{-1}]$$

where:

$BD$—bulk density [Mg·m$^{-3}$]
$OM$—organic matter content [%],
$C$—clay fraction content [%],
$S_i$—silt fraction content [%].

**Method 14—MRA (multiple regression analysis).**
Model of MRA was carried out based on data published by Ryczek et al. (2017)

$$K_s = 3.61216 + 0.04474 \cdot S + 0.01300 \cdot S_i - 2.42722 \cdot C - 3.28861 \cdot n \; [\text{m} \cdot \text{day}^{-1}]$$

where:

$S$—sand fraction (2–0.05 mm) content [%],
$S_i$—silt fraction (0.05–0.002 mm) content [%],
$C$—clay fraction (<0.002 mm) content [%],
$n$—as above [-].

Regression coefficients were calculated in Statistica program release 13.5.

**Method 15—ANN (Artificial Neural Networks)** [43,64,65].

In the work we used the ANN model MLP 11-11-1, as described by [66]. The input data were 11 soil parameters: content of clay, silt and sand fractions, as well as total porosity, organic matter content and effective diameters: d10, d20, d50, d60 i d90, and bulk and solid phase density.

*2.2. Soil Properties*

The methods for determination of soil properties were presented in Table 1.

**Table 1.** Methods for determination of soil properties.

| Soil Property | Methods |
|---|---|
| texture | the Casagrandesedimentation and sieve methods; classification of fractions and granular groups was carried out according to USDA (United States Department of Agriculture) |
| total porosity ($n$) | $n = 1 - BD \cdot SD^{-1}$, where: $BD$—bulk density, measured by means of ring method, $SD$—specific density, measured by means of pycnometric method |
| void ratio ($e$) | $e = n \left(1 - n\right)^{-1}$ |
| saturated hydraulic conductivity ($K_s$) | doublering infiltrometer |
| organic matter content ($OM$) | the Tiurin method |

*2.3. Statistical Analysis*

The adjustment of the results obtained by means of the chosen pedotransfer functions to the ones obtained using the double ring method was evaluated by means of some statistical parameters, as [40,67]:

- mean error of prognosis (*MEP*)

$$MEP = \frac{1}{n} \cdot \sum_{i=1}^{n} \left( C_i^m - C_i^p \right)$$

- root of mean square error (*RMSE*)

$$RMSE = \sqrt{\frac{1}{n} \cdot \sum_{i=1}^{n} \left( C_i^m - C_i^p \right)^2}$$

- mean percentage error (*MPE*)

$$MPE = \frac{1}{n} \cdot \sum_{i=1}^{n} \frac{C_i^m - C_i^p}{C_i^m} \cdot 100$$

- model efficiency (*ME*) [41,45]

$$ME = 1 - \frac{\sum_{i=1}^{n} \left( C_i^m - C_i^p \right)^2}{\sum_{i=1}^{n} \left( C_i^m - \overline{C} \right)^2}$$

where:

$C_i^m$—measured values,
$C_i^p$—simulated values,
$n$—number of data,
$\overline{C}$—mean measured value.

Statistical significance of differences between pedotransfer functions were checked by means of $LSD_{Tukey}$ (least significant differences by Tukey's test).

### 2.4. Investigated Site

The field experiment was carried out on the evidence plot 647, precint Brzeźnica, evidence unit Rudnik (community), Silesia voivodship, Racibórz district (Figure 1), belonging to the Agriculture-Industry Enterprise in Racibórz, LC. The experiment site belongs to the mesoregion Racibórz Valley [68,69]. According to the Gumiński agricultural-climatic provinces, the site belongs to province Sub Sudety—XVIII. Samples were taken from 9 points, located in regular squares network (Figure 2). The investigated site is characterized by high slope attaining 20°. It has been used as arable land, under maize and earlier under winter wheat. According to texture (15% of sand, 75% of silt, 10% of clay), soil is classified as: silt loam (SiL) [53]. It is the typical loamy loess and undergoes high water erosion.

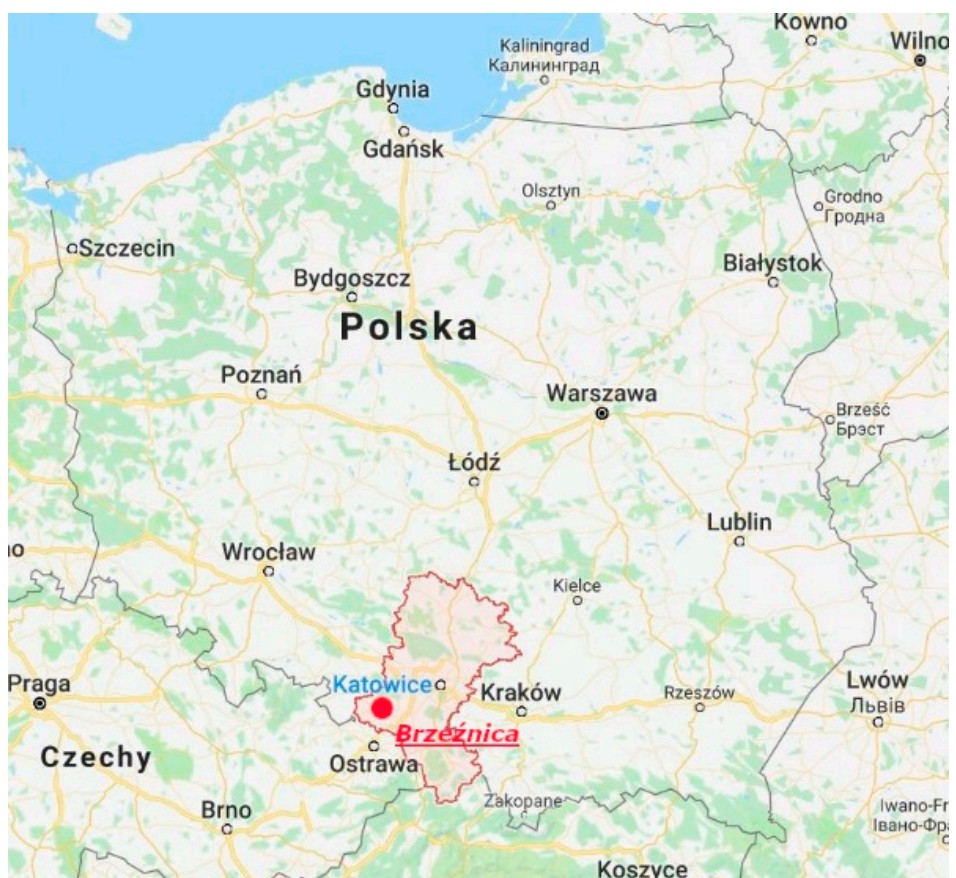

**Figure 1.** Location of the investigated site (source: www.google.pl/maps (accessed on 20 November 2022)).

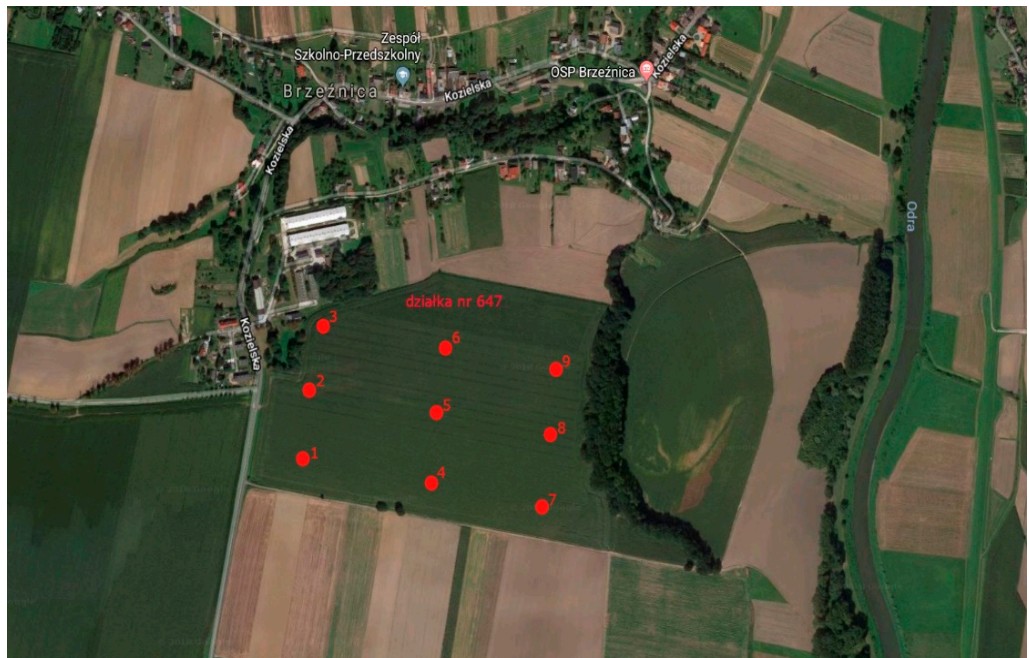

**Figure 2.** Location of experiment points (source: www.googlemap.pl (accessed on 20 November 2022)).

## 3. Results and Discussion

In Table 2 there are presented values of some statistical measures of soil parameters used for calculation of saturated hydraulic conductivity by use of pedotransfer functions. Regarding texture, soil on the investigated site is classified as silt loam (SiL). Effective diameter $d_{10}$ varied from $1.7 \cdot 10^{-3}$ to $3.0 \cdot 10^{-3}$ mm, $d_{20}$ between $4.5 \cdot 10^{-3}$ and $8.0 \cdot 10^{-3}$ mm, while $d_{50}$ between $2.8 \cdot 10^{-3}$ mm and $3.8 \cdot 10^{-2}$ mm. Values of organic matter fluctuated between 0.85 and 1.35%, while total porosity between 0.394 and 0.481. Bulk density attained values between 1.41 and 1.57 Mg·m$^{-3}$. Values of saturated hydraulic conductivity K$_s$ for analyzed points fluctuated between $3.25 \cdot 10^{-2}$ and $8.72 \cdot 10^{-2}$ m·day$^{-1}$.

**Table 2.** Statistical values of parameters for determination of saturated hydraulic conductivity by means of thepedotransfer functions.

| Soil Parameter | Statistical Parameters | | | | |
|---|---|---|---|---|---|
| | $V_{min}$ | $V_{max}$ | $\bar{x}$ | $\sigma_{n-1}$ | $V$ (%) |
| $C$ [%] | 5 | 12 | 10 | 2 | 20.0 |
| $S_i$ [%] | 70 | 79 | 75 | 3 | 4.0 |
| $S$ [%] | 13 | 18 | 15 | 2 | 13.3 |
| $d_{10}$ [mm] | $1.7 \cdot 10^{-3}$ | $3.0 \cdot 10^{-3}$ | $2.1 \cdot 10^{-3}$ | $5.0 \cdot 10^{-4}$ | 23.7 |
| $d_{20}$ [mm] | $4.5 \cdot 10^{-3}$ | $8.0 \cdot 10^{-3}$ | $6.2 \cdot 10^{-3}$ | $1.2 \cdot 10^{-3}$ | 19.4 |
| $d_{50}$ [mm] | $9.6 \cdot 10^{-3}$ | $2.8 \cdot 10^{-2}$ | $2.2 \cdot 10^{-2}$ | $5.0 \cdot 10^{-3}$ | 22.7 |
| $d_{60}$ [mm] | $2.8 \cdot 10^{-3}$ | $3.8 \cdot 10^{-2}$ | $2.5 \cdot 10^{-2}$ | $1.0 \cdot 10^{-2}$ | 40.0 |
| $d_{90}$ [mm] | $6.0 \cdot 10^{-2}$ | $8.5 \cdot 10^{-2}$ | $7.3 \cdot 10^{-2}$ | $8.2 \cdot 10^{-3}$ | 11.2 |
| $d_e$ [mm] | $1.1 \cdot 10^{-2}$ | $1.8 \cdot 10^{-2}$ | $1.3 \cdot 10^{-2}$ | $2.2 \cdot 10^{-3}$ | 16.8 |
| $a$ [-] | 0.02 | 0.06 | 0.04 | 0.01 | 25.0 |
| $OM$ [%] | 0.85 | 1.35 | 1.06 | 0.15 | 13.9 |
| $n$ [-] | 0.394 | 0.481 | 0.433 | 0.031 | 7.1 |
| $e$ [-] | 0.650 | 0.927 | 0.769 | 0.097 | 12.6 |
| $BD$ [Mg·m$^{-3}$] | 1.41 | 1.57 | 1.50 | 0.06 | 4.1 |
| $K_s$ [m·day$^{-1}$] | $3.25 \cdot 10^{-2}$ | $8.72 \cdot 10^{-2}$ | $6.59 \cdot 10^{-2}$ | $1.96 \cdot 10^{-2}$ | 29.7 |

Meanings: $C$ —clay fraction (<0.002 mm) content, $S_i$ —silt fraction (0.05–0.002 mm) content, $S$ —sand fraction (2–0.05 mm) content, $d_e$ —proper effective diameter [mm], $a$ —content of particles below 0.001 mm, $OM$ —organic matter content, $n$ -total porosity, $e$ —void ratio, $BD$ —bulk density, $K_s$ —saturated hydraulic conductivity.

Table 3 presents some statistical measures for saturated hydraulic conductivity determined by one of the fifteen pedotransfer functions. Generally, the obtained mean values were between $9.10 \cdot 10^{-4}$ and $4.59 \cdot 10^{0}$ m·day$^{-1}$.

**Table 3.** Statistical parameters of the saturated hydraulic conductivity obtained for the chosenpedotransfer functions.

| Method | Statistical Parameters | | | | |
| | $V_{min}$ | $V_{max}$ | $\bar{x}$ | $\sigma_{n-1}$ | $V$ |
| | (m·day$^{-1}$) | | | | (%) |
|---|---|---|---|---|---|
| 1 | $2.71 \cdot 10^{-3}$ | $9.83 \cdot 10^{-3}$ | $5.04 \cdot 10^{-3}$ | $2.50 \cdot 10^{-3}$ | 49.6 |
| 2 | $6.45 \cdot 10^{-3}$ | $1.69 \cdot 10^{-1}$ | $4.22 \cdot 10^{-2}$ | $5.22 \cdot 10^{-2}$ | 123.6 |
| 3 | $1.25 \cdot 10^{-3}$ | $4.68 \cdot 10^{-3}$ | $2.73 \cdot 10^{-3}$ | $1.15 \cdot 10^{-3}$ | 42.1 |
| 4 | $1.21 \cdot 10^{-2}$ | $2.22 \cdot 10^{-2}$ | $1.59 \cdot 10^{-2}$ | $3.23 \cdot 10^{-3}$ | 20.3 |
| 5 | $2.62 \cdot 10^{-2}$ | $1.24 \cdot 10^{-1}$ | $6.51 \cdot 10^{-2}$ | $3.28 \cdot 10^{-2}$ | 50.4 |
| 6 | $4.36 \cdot 10^{-2}$ | $1.27 \cdot 10^{-1}$ | $7.71 \cdot 10^{-2}$ | $2.67 \cdot 10^{-2}$ | 34.6 |
| 7 | $1.10 \cdot 10^{-2}$ | $4.16 \cdot 10^{-2}$ | $2.14 \cdot 10^{-2}$ | $1.06 \cdot 10^{-2}$ | 49.5 |
| 8 | $1.83 \cdot 10^{-3}$ | $1.15 \cdot 10^{-2}$ | $4.90 \cdot 10^{-3}$ | $4.11 \cdot 10^{-3}$ | 83.9 |
| 9 | $8.91 \cdot 10^{-2}$ | $4.45 \cdot 10^{-1}$ | $2.54 \cdot 10^{-1}$ | $9.57 \cdot 10^{-2}$ | 37.6 |
| 10 | $6.20 \cdot 10^{-4}$ | $2.34 \cdot 10^{-2}$ | $5.87 \cdot 10^{-3}$ | $7.28 \cdot 10^{-3}$ | 123.9 |
| 11 | $2.72 \cdot 10^{-2}$ | $1.24 \cdot 10^{-1}$ | $7.05 \cdot 10^{-2}$ | $3.07 \cdot 10^{-2}$ | 43.5 |
| 12 | $9.10 \cdot 10^{-4}$ | $3.87 \cdot 10^{-3}$ | $2.00 \cdot 10^{-3}$ | $9.90 \cdot 10^{-4}$ | 49.4 |
| 13 | $3.70 \cdot 10^{0}$ | $4.59 \cdot 10^{0}$ | $4.05 \cdot 10^{0}$ | $3.14 \cdot 10^{-1}$ | 7.7 |
| 14 | $2.63 \cdot 10^{-2}$ | $4.74 \cdot 10^{-1}$ | $2.05 \cdot 10^{-1}$ | $1.28 \cdot 10^{-1}$ | 62.7 |
| 15 | $5.20 \cdot 10^{-2}$ | $5.21 \cdot 10^{-2}$ | $5.21 \cdot 10^{-2}$ | $3.00 \cdot 10^{-5}$ | 0.1 |

The analysis of variance was introduced in Table 4, while Table 5 presents statistically uniform groups regarding statistical essentiality. Method 13 (Fournivaland Wilson) showed statistically essential difference in relation to the other methods. In turn, between method 9 (Seelheim) and 14 (multiple regression) there is no statistically essential difference. Between method 14 (multiple regression) and the remaining following methods there is no statistical essential difference.

**Table 4.** Analysis of variance.

| Variability Source | Squares Sum | Freedom Degrees | Mean Square |
|---|---|---|---|
| Points | 0.111 | 8 | |
| Methods | 134.509 | 14 | 9.608 |
| Error | 0.917 | 112 | 0.008 |
| Total | 135.537 | 134 | |

Calculationof the LSD by the Tukey test: the Tukey distribution q for $\alpha = 0.05$, $\nu = 112$ and m = 15 is equal 4.91, standard deviation of arithmetic mean: $s_x = 0.03016$. LSD = 0.148.

In Table 6 there are presented results of statistical analysis of comparison of results obtained by means of field direct methods with the ones obtained by means of the chosen pedotransfer methods. The results show that the method 1 (Hazen), 3 (USBR), 4 (Saxton), 7 (Tezaghi), 8 (Chapuis), 9 (Seelheim), 10 (Chapuis-NAVFAC, 12 (Slichter), 13 (Furnival) and 14 (multiple regression) gave statistically essential differences in relation to the results obtained by means of the field method. Comparing, in turn, percentage differences, the least one showed method 5 (Kozeny-Carman, underestimation attaining 1.4%), and the highest showed method 13 (Furnival and Wilson, overestimation attained as many as 6036.4%).

**Table 5.** Analysis of essential differences between results of determination by means of thepedotransfer functions.

| Method | Mean [m·day$^{-1}$] | Homogenous Groups |
|--------|---------------------|-------------------|
| 13 | $4.05 \cdot 10^{0}$ | a |
| 9 | $2.54 \cdot 10^{-1}$ | b |
| 14 | $2.05 \cdot 10^{-1}$ | bc |
| 6 | $7.70 \cdot 10^{-2}$ | c |
| 11 | $7.05 \cdot 10^{-2}$ | c |
| 5 | $6.51 \cdot 10^{-2}$ | c |
| 15 | $5.21 \cdot 10^{-2}$ | c |
| 2 | $4.22 \cdot 10^{-2}$ | c |
| 7 | $2.14 \cdot 10^{-2}$ | c |
| 4 | $1.59 \cdot 10^{-2}$ | c |
| 10 | $6.92 \cdot 10^{-3}$ | c |
| 1 | $5.04 \cdot 10^{-3}$ | c |
| 8 | $3.39 \cdot 10^{-3}$ | c |
| 3 | $2.73 \cdot 10^{-3}$ | c |
| 12 | $2.00 \cdot 10^{-3}$ | c |

Mean value of saturated hydraulic conductivity obtained by means of double-ring method was: $6.59 \cdot 10^{-2}$ m·day$^{-1}$ with standard deviation $1.96 \cdot 10^{-2}$ m·day$^{-1}$, and variation coefficient 29.7%.

**Table 6.** Values of t-Student test between results obtained by means of the pedotransferfunctionswith the ones obtained by means of measured data.

| Method | Test t-Student Value | Critical Value t$_{0.05}$ | Difference % ** |
|--------|----------------------|---------------------------|-----------------|
| 1 | −10.567 * | | 92.4 |
| 2 | −0.743 | | 36.1 |
| 3 | −11.273 * | | 95.0 |
| 4 | −8.477 * | | 75.9 |
| 5 | −0.030 | | 1.4 |
| 6 | 0.677 | | −16.7 |
| 7 | −5.278 * | | 67.6 |
| 8 | −11.061 * | | 94.9 |
| 9 | 3.286 * | | −284.8 |
| 10 | −8.396 * | | 89.5 |
| 11 | 0.251 | 2.228 | −6.8 |
| 12 | −11.426 * | | 97.0 |
| 13 | 21.423 * | | −6036.4 |
| 14 | −2.452 * | | −210.6 |
| 15 | 1.812 | | 21.1 |

* differences are statistically essential, ** positive values show underestimation of the pedotransfer function in relation to the measured values, while negative values show overestimation.

For the purpose to choose the best function simulating saturated hydraulic conductivity for the loess soil, the various model efficiency measures were used (Table 7). Values of correlation coefficient *r* for pedotransfer functions fluctuated between 0.059 and 0.708.Only for methods 2nd (Hazen-Tkaczukowa), 4th (Saxton), 10th (NAVFAC) and 15th (ANN) coefficients were statistically essential for confidence level0.01.The best accordance with the field double-ring method regarding correlation coefficient had the 15th (ANN) method, while the most abandoning ones were 11th (Sauerbrej) and 12th (Slichter) functions. Results obtained for *MEP* showed that maximum underestimation attained $2.18 \cdot 10^{-2}$, for 15th (ANN) function, while little overestimation took place for 11th (Sauerbrej) function. Mean percentage error *MPE* shows good results of estimation for 2nd (Hazen-Tkaczukowa) function. Its value was 6.2%. Extreme bad adjustment had 13th (Furnival and Wilson) (as many as −5774.6%) function. Root of mean square error *RMSE* attained the highest value for 13th (Furnival and Wilson) function($3.99 \cdot 10^{0} \cdot 10^{0}$ m·day$^{-1}$). The best results attained 15th (ANN) function. Analysis of homogeneity of mean values (Table 4) using the Tukey's

test $LSD_{Tukey}$ showed that the 13th (Furnival and Wilson) method differed statistically in comparison to other functions (Table 8). Functions 9th (Seelheim) and 14th (MRA) did not differ between them and differed statistically from the remaining methods.

**Table 7.** Model efficiency measures.

| Model | Efficiency Measures | | | | |
| --- | --- | --- | --- | --- | --- |
| | *MEP* [m·day$^{-1}$] | *RMSE* [m·day$^{-1}$] | *MPE* [%] | *ME* [-] | *r* [-] |
| 1 | $6.09 \cdot 10^{-2}$ | $6.40 \cdot 10^{-2}$ | 90.8 | $-10.973$ | 0.440 |
| 2 | $2.37 \cdot 10^{-2}$ | $6.69 \cdot 10^{-2}$ | 6.2 | $-12.070$ | 0.630 * |
| 3 | $6.32 \cdot 10^{-2}$ | $6.60 \cdot 10^{-2}$ | 94.9 | $-11.743$ | 0.501 |
| 4 | $5.00 \cdot 10^{-2}$ | $5.42 \cdot 10^{-2}$ | 71.6 | $-7.577$ | 0.716 * |
| 5 | $8.22 \cdot 10^{-4}$ | $3.94 \cdot 10^{-2}$ | $-14.2$ | $-3.540$ | 0.221 |
| 6 | $-1.11 \cdot 10^{-2}$ | $3.88 \cdot 10^{-2}$ | $-38.3$ | $-3.414$ | 0.439 |
| 7 | $4.46 \cdot 10^{-2}$ | $5.08 \cdot 10^{-2}$ | 61.2 | $-6.547$ | 0.416 |
| 8 | $6.25 \cdot 10^{-2}$ | $6.54 \cdot 10^{-2}$ | 94.0 | 11.496 | 0.300 |
| 9 | $-1.88 \cdot 10^{-1}$ | $2.12 \cdot 10^{-1}$ | $-355.5$ | $-131.000$ | 0.361 |
| 10 | $5.51 \cdot 10^{-2}$ | $5.73 \cdot 10^{-2}$ | 85.1 | $-8.585$ | 0.549 * |
| 11 | $-4.54 \cdot 10^{-3}$ | $3.55 \cdot 10^{-2}$ | $-22.4$ | $-2.692$ | 0.059 |
| 12 | $7.19 \cdot 10^{-2}$ | $7.40 \cdot 10^{-2}$ | 97.2 | $-13.436$ | 0.059 |
| 13 | $-3.98 \cdot 10^{0}$ | $3.99 \cdot 10^{0}$ | $-5774.6$ | $-41,907.886$ | 0.156 |
| 14 | $-1.31 \cdot 10^{-1}$ | $1.77 \cdot 10^{-1}$ | $-185.5$ | 81.138 | 0.206 |
| 15 | $2.18 \cdot 10^{-2}$ | $2.81 \cdot 10^{-2}$ | 24.0 | $-1.080$ | 0.708 * |

*—statistically essential for confidence level 0.1.

**Table 8.** Values of parameters of spatial distribution.

| Methods | Mean Value | Standard Deviation | Variability Coefficient |
| --- | --- | --- | --- |
| 1 | $5.04 \cdot 10^{-3}$ | $2.50 \cdot 10^{-3}$ | 49.7 |
| 2 | $4.22 \cdot 10^{-2}$ | $5.21 \cdot 10^{-2}$ | 123.6 |
| 3 | $2.73 \cdot 10^{-3}$ | $1.15 \cdot 10^{-3}$ | 42.1 |
| 4 | $1.59 \cdot 10^{-2}$ | $3.24 \cdot 10^{-3}$ | 20.3 |
| 5 | $6.51 \cdot 10^{-2}$ | $3.27 \cdot 10^{-2}$ | 50.3 |
| 6 | $7.70 \cdot 10^{-2}$ | $2.66 \cdot 10^{-2}$ | 34.5 |
| 7 | $2.14 \cdot 10^{-2}$ | $1.06 \cdot 10^{-2}$ | 49.4 |
| 8 | $3.39 \cdot 10^{-3}$ | $1.68 \cdot 10^{-3}$ | 49.4 |
| 9 | $2.54 \cdot 10^{-1}$ | $9.58 \cdot 10^{-2}$ | 37.7 |
| 10 | $6.92 \cdot 10^{-3}$ | $7.17 \cdot 10^{-3}$ | 103.6 |
| 11 | $7.05 \cdot 10^{-2}$ | $3.07 \cdot 10^{-2}$ | 43.5 |
| 12 | $2.00 \cdot 10^{-3}$ | $9.90 \cdot 10^{-4}$ | 49.4 |
| 13 | $4.05 \cdot 10^{0}$ | $3.12 \cdot 10^{-1}$ | 7.7 |
| 14 | $2.05 \cdot 10^{-1}$ | $1.28 \cdot 10^{-1}$ | 62.7 |
| 15 | $5.21 \cdot 10^{-2}$ | $2.07 \cdot 10^{-5}$ | 0.0 |

## 4. Conclusions

1. The LSD analysis showed that the Fournival and Wilson method, based on texture and total porosity differs statistically for investigated site from the other methods. In turn, between the Seelheim (based on texture only) and multiple regression methods there is not a statistical difference. Between the Seelheim method and the other ones there are not a statistically essential difference;
2. The t-Student analysis showed that the methods: Hazen, USBR, Saxton, Seelheim (based only on texture), Chapuis, NAVFAC, Furnival and Wiliam, multiple regression (based on texture and total porosity), and Slichter and Tezaghi (based on texture, total porosity and water properties)gave statistically essential differences in comparison to the results obtained by the field method. The remaining method does not differ statistically from the field method;

3. Comparing the percentage differences, the lowest showed the Kozeny-Carman method, in which underestimation in relation to the field method was 1.4%. In turn, the highest difference was in the case of the Furnival and Wilson, in which overestimation was as many as 5.4%. The lowest differences were in the case of the methods where total porosity was taken into account;

4. The highest spatial variability was for the Hazen and Tkaczukowa methods, where variability coefficient was 123.6%. In turn, the artificial neural network method was characterized by a lack of variability.

**Author Contributions:** Conceptualization, A.P. and M.R.; methodology, A.P. and M.R.; validation, E.K., L.L.; formal analysis, M.R.; resources M.R. and E.K.; writing—original draft preparation, M.R. and A.P.; writing—review and editing, A.P. and M.R.; supervision, E.K. and L.L.; project administration, A.P.; funding acquisition, A.P. and E.K. All authors have read and agreed to the published version of the manuscript.

**Funding:** The publication was co-financed from the subsidy granted to the Cracow University of Economics—Project nr 28/GGR/2021/POT.

**Institutional Review Board Statement:** Not applicable.

**Informed Consent Statement:** Not applicable.

**Conflicts of Interest:** The authors declare no conflict of interest.

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
