# Peer review of "Comparison of Pedotransfer Functions for Determination of Saturated Hydraulic Conductivity for Highly Eroded Loess Soil"

_land, doi:10.3390/land12030610_

Round 1

Reviewer 1 Report

The work compared 15 pedotransfer models and determination of their usefulness for assessment of Saturated Hydraulic Conductivity for highly eroded loess soil. Saturated Hydraulic Conductivity varies greatly, and there are many influencing factors. However, the manuscript barely introduces these aspects.The format needs to be adjusted. Many formulas are too small, and the parameter descriptions are not uniform. The font size of title of table 6 is not right. It is suggested that the article be improved.

L12: This sentence is meaningless. Almost all experiments are time or cost-consuming.

L27: there is too many references in one sentence. It is recommended to explain these.

L32: These cited articles are generally early. It is recommended to retrieval the latest references.

[1]   Analysis of Temperature Effect on Saturated Hydraulic Conductivity of the Chinese Loess. Doi: 10.3390/w14091327

[2] Pedotransfer functions developed for calculating soil saturated hydraulic conductivity in check dams on the Loess Plateau in China. Doi: 10.1002/vzj2.20217

L55: d-1 ?It shall be unified with other Ks units.

L56: Ks also needs to be described.

L65: d10 is no need to reintroduce.

L173, L175: the formulas need to be rewrited.

Table 2: For the same parameter, the valid digits must be consistent.

L250: ANN is a best matching algorithm, and it is inevitable that the effect will be good. Is it meaningful to compare with other methods?

Author Response

Responses to remarks contained in review 1

L27: there is too many references in one sentence. It is recommended to explain these.

It was improved.

L32: These cited articles are generally early. It is recommended to retrieval the latest references.

It was supplemented.

[1]   Analysis of Temperature Effect on Saturated Hydraulic Conductivity of the Chinese Loess. Doi: 10.3390/w14091327

[2] Pedotransfer functions developed for calculating soil saturated hydraulic conductivity in check dams on the Loess Plateau in China. Doi: 10.1002/vzj2.20217

It was supplemented.

L55: d-1 ?It shall be unified with other Ks units.

It was improved.

L56: Ks also needs to be described.

It was improved.

L65: d10 is no need to reintroduce.

It was improved.

Table 2: For the same parameter, the valid digits must be consistent.

It was improved.

Thank You very much for very precious remarks.

Reviewer 2 Report

Manuscript notes

- Chapter „Introduction” should be supplemented by informations concerning pedotransfer functions and significance of saturated hydraulic conductivity in investigations of erosion

- In description of method 5,   - specific density should be expressed in [MgË‘m-3]

- In description of method 7, d10 should be expressed in [mm]

- In description of Method 12 please write, what value of m were taken (what functions were used)

- In description of method 13 bulk density should be expressed in [Mgˑm-3]

- Some references describing Methods should be newer

- There lacks item [Kruk et al. 2017]

- In line 13 counting at end of the Chaper Results and Discussion) should be (with the field method).

Author Response

Responses to remarks contained in review 2

- Chapter „Introduction” should be supplemented by informations concerning pedotransfer functions and significance of saturated hydraulic conductivity in investigations of erosion

It was supplemented.

- In description of method 5,   - specific density should be expressed in [MgË‘m-3]

It was improved.

- In description of method 7, d10 should be expressed in [mm]

It was improved.

- In description of Method 12 please write, what value of m were taken (what functions were used)

It was supplemented.

- In description of method 13 bulk density should be expressed in [Mgˑm-3]

It was improved.

- Some references describing Methods should be newer

It was improved and some itmes were added.

- There lacks item [Kruk et al. 2017]

It was improved.

- In line 13 counting at end of the Chaper Results and Discussion) should be (with the field method).

It was improved.

Thank You very much for very precious remarks.

Reviewer 3 Report

Summary:

 In this paper the authors carried out in field permeability tests and the determined values of the hydraulic conductivity were compared with those derived by the application of 15 empirical methods. The proposed research is interesting for the great potential of predictive models. The objectives of the paper are well defined and presented in a logical order and the discussion is clear and consistent with the obtained results. For the above reasons I recommend the publication of this paper if the general/major comments and the specific ones are taken into account.

General/Major Comments:

It is necessary to define the following aspects:

1) Make sure that the empirical relationships proposed in the literature can be applied to the studied soils. For example the Navfac method provides Ksat under five conditions which are defined in the paper: Chapuis, R.P. Predicting the saturated hydraulic conductivity of soils: A review. Bull. Eng. Geol. Environ. 2012, 71, 401–434.

Please define, if they exist, the conditions of validity of the various methods in the text and therefore make it explicit that the proposed methods are valid for the studied soils.

2) Pay attention to formatting (e.g. references in the text are to be cited by numbers) and use “.” for decimals.

3) Replace “Kozena” with “Kozeny”

4) Note that some expressions are not completely empirical. For example the Kozeny-Carman equation is derived by semiempirical and theoretical evaluations (the following references can be useful: Chapuis 2012 Predicting the saturated hydraulic conductivity of soils: A review; Mbonimpa et al 2022, Practical pedotransfer functions for estimating the saturated hydraulic conductivity; Bilardi et al 2021 Predicting the Saturated Hydraulic Conductivity of Clayey Soils and Clayey or Silty Sands; Sanzeni et al 2013 Specific Surface and Hydraulic Conductivity of Fine-Grained Soils).

5) Add considerations about why some methods work better than others with reference to the parameters taken into account by the different methods, probably some parameters are more important than others.

Specific comments:

Page 2 line 56: Please pay attention to units of measurement (i.e. “d” should be replaced with “day”)

Page 2 line 58: Define D10 as done in line 66

Page 2 line 65: Please define units of measurement of d<0.001

Page 2 line 80: Check the Kozeny – Carman equation (i.e. CKC multiply S)

Page 2 line 85: How was the specific surface area calculated?

Page 3 line 97: Please define t in the formula of Terzaghi

Page 3 line 121: Specify that this formula is valid for: 0.3 < e < 0.7 (see also general comments)

Page 4 line 136: What value of m was used?     

Page 7 Table 2: Check the units of measurement of “The content of particles below 0,001 mm or ?<0,001”

Page 7 line 198: Please add units of measurement after 0.008

Page 7 line 205: Clay fraction (<0,02 mm) should be <0.002 mm

Page 7 lines 206-207: de is define two times but it is unclear what it represents or how it was determined

Page 7: The minimum vale of bulk density is 1.47 in the text (line 201) and 1.41 in Table 1

Author Response

Responses to remarks contained in review 2

L27: thereistoomanyreferences in one sentence. It isrecommended to explainthese.

It was improved.

L32: Thesecitedarticlesaregenerallyearly. It isrecommended to retrieval the latestreferences.

It was supplemented.

[1]   Analysis of TemperatureEffect on SaturatedHydraulicConductivity of the ChineseLoess. Doi: 10.3390/w14091327

[2] Pedotransferfunctionsdeveloped for calculatingsoilsaturatedhydraulicconductivity in checkdams on the Loess Plateau in China. Doi: 10.1002/vzj2.20217

It was supplemented.

L55: d-1 ?It shall be unified with otherKsunits.

It was improved.

L56: Ksalsoneeds to be described.

It was improved.

L65: d10 is no need to reintroduce.

It was improved.

Table 2: For the same parameter, the valid digits must be consistent.

It was improved.

 Thank You very much for very precious remarks.

Round 2

Reviewer 1 Report

The introduction is too brief and does not clearly show the existing research results.

Author Response

All the minor remarks were regarded.

Reviewer 3 Report

Lines 91, 236 and 277:Replace “Kozena” with “Kozeny”

Author Response

All the minor remarks were regarded.